# Who benefits from the donor-supported malaria programme in Enugu State, Nigeria? A benefit incidence analysis

Eric Obikeze[1,2], Wenhui Mao[3]*, Uchenna Ezenwaka[1,2], Ifeyinwa Arize[1,2], Osondu Ogbuoji[3], Obinna Onwujekwe[1,2]

1 Department of Health Administration and Management, Faculty of Health Sciences and Technology, College of Medicine, University of Nigeria Nsukka (Enugu Campus), Enugu, Nigeria, 2 Health Policy Research Group, Department of Pharmacology and Therapeutics, College of Medicine, University of Nigeria Nsukka (Enugu-Campus), Enugu Nigeria, 3 The Center for Policy Impact in Global Health, Duke Global Health Institute, Duke University, Durham, North Carolina, United States of America

* wenhui.mao@duke.edu

## Abstract

Nigeria bears the highest global burden of malaria, accounting for 25% of cases and 19% of deaths worldwide. Development partners provide substantial support for malaria prevention and treatment in Nigeria. This study examines the financial burden of malaria on households and the benefit incidence of donor-supported bed net services in Enugu State, Nigeria. We conducted an interview-administered household survey in urban, semi-urban and rural regions in Enugu State in 2020. We collected data on the use of malaria services and out-of-pocket (OOP) payments. Socioeconomic status (SES) was estimated using household assets ownership. The benefits of malaria services were calculated by multiplying the unit cost of services while the net benefit was calculated by subtracting OOP payment from the benefits. A concentration index was used to assess equity in spending on malaria across socioeconomic quintiles. We estimated the gross and net benefit incidences for malaria services by deducting the OOP payment from the gross benefits. Most respondents were women, married, and had attained secondary education. Over 53.9% of surveyed households owned bed net. About 31.6% of households used malaria drugs in the past months. All users paid OOP for malaria drugs, sprays and lab services and over one-third of households incurred OOP costs for bed nets. The total OOP expenditure for malaria in the past month was $0.53 per household. The gross benefit incidence for malaria services was $1836.7. The net benefit and donor benefit were $1679.5 and $705.4, respectively. Both gross and net benefit for malaria services favored less-poor households. Households in Enugu State incur OOP expenses for malaria diagnosis and treatment, and less-poor households benefit more from government- and donor- subsidized malaria services, including bed nets. It is imperative to improve the accessibility and affordability of malaria diagnosis and treatment in Nigeria to ensure equitable access to malaria services.

**Data availability statement:** Household survey, including demographic, health service use, and household expenditure information, has been collected to perform this research. On our IRB protocol and the informed consent form, we did not propose for data sharing. Therefore, data will not be publicly available. However, a de-identified dataset could be shared for research purpose upon request. Please email corresponding author or cpighcoreteam@duke.edu with your study proposal for data request.

**Funding:** This paper was part of the project "Driving health progress during disease, demographic, domestic finance and donor transitions (the "4Ds"): policy analysis and engagement with six transitioning countries" funded by Bill and Melinda Gates Foundation (OPP1199624). The funders had no role in study design, data collection and analysis, decision to publish, or preparation of the manuscript.

**Competing interests:** The authors have declared that no competing interests exist.

## Introduction

Malaria is a significant public health challenge in 97 countries and territories [1]. Nigeria bears a disproportionate burden of malaria, with an estimated 100 million clinically diagnosed cases and 95,000 deaths in 2018 [2], accounting for about one-fourth of malaria cases and one-fifth of malaria deaths globally [3]. Malaria transmission is unevenly distributed across the country [4] with individuals of lower economic status being more vulnerable to infections and less likely to access proper care [5].

Malaria places a considerable strain on Nigeria's health system and households. Over 40% of outpatient consultations, 30% of hospitalizations, and 11% of maternal mortalities in Nigeria were attributed to malaria in 2018. It accounts for over 40% of the total monthly curative healthcare costs incurred by households, surpassing the combined costs of other illnesses [6,7,8].

Nigeria's fragile health infrastructure and inadequate healthcare financing strategies compel the population to pay out of pocket (OOP) for malaria diagnosis and treatment [9], with the exception of a few donor-funded programmes. The financial burden of medical and related costs presents a significant barrier to seeking proper care for malaria [10].

In Nigeria, free access to malaria care in public facilities is available only in 12 out of 36 states, including Enugu State, where free maternal and child health are provided [11,12]. Malaria services are predominantly funded by development partners. In 2021, the Enugu state government's expenditure on long-last insecticide-treated nets (LLINs) was a mere 0.07% of the total health budget, amounting to N10,660,000 million [13]. In contrast, development partners contributed over 99% of the funding for malaria. From 2014 to 2017, donor financing for malaria in Nigeria totaled $1.15 billion with Global Fund being the largest contributor ($179.4 million) [14]. The US President's Malaria Initiative (PMI), launched in 2005, aimed to reduce malaria-related mortality by 50% across 15 high-burden countries, including Nigeria [15]. Four key approaches were implemented by PMI across 11 states (with Enugu State added as a non-PMI focus site): 1) insecticide-treated mosquito nets (ITNs); 2) indoor residual spraying (IRS); 3) accurate diagnosis and prompt treatment with artemisinin-based combination therapies (ACTs); and 4) intermittent preventive treatment of pregnant women (IPTp) [16,17].

Enugu state has allocated varying budgets to support different malaria control programs in recent years. In 2020, Enugu state budgeted N30,000,000 for malaria control, covering training, procurement of long-lasting insecticide-treated nets (LLINs), drugs, test kits, chemicals and equipment, and social mobilization efforts. In 2021, only N10 million was allocated for the procurement of LLINs. In 2022, another N10million was budgeted for malaria elimination programme, which included the procurement of LLINs, mapping and geographic reconnaissance for IRS areas. For 2023, the budget increased to N17million for the procurement and delivery of ten thousand (10,000) LLINs to pregnant women and children under 5 as well as the procurement of ten thousand (10,000) rapid diagnostic tests (RDTs) [18].

Development partners and domestic government have provided financial resources, medical commodities and other support to enhance malaria care and reduce the financial burden on households in Nigeria [19,20]. While there is ample evidence regarding the inputs, process and outputs of these malaria initiatives--such as investment amounts and the number of bed nets delivered [21] --little is known about the benefit incidence of these programs.

However, the provision of malaria services does not always result in equitable outcomes for households in Nigeria, particularly for the poor and vulnerable. As Nigeria transitions away from donor assistance, concerns arise about sustaining the progress made in malaria control. while the Global Fund plans to remain in Nigeria beyond 2027, and GAVI has renewed its support through National Primary Health Care Development Agency (NPHCDA) [22],

the challenge remains in determining how the domestic government can take over malaria programs and minimize the negative impact of donor transition. Understand the benefit incidence of donor-funded malaria programs. This paper aims to assess which households have benefited the most from these programs. This will ensure better prioritization of health care commodities and more effective allocation of domestic resources to sustain the health gains after donor transition.

## Materials and methods

### Study sites

The study was conducted in three Local Government Areas (LGAs) in Enugu state, southeast Nigeria. Enugu State, with its capital city, Enugu, is made up of seventeen LGAs, five of which are urban, while the remaining are rural. The state has an estimated population of 4.5 million people [23]. There are six district hospitals, 36 cottage hospitals and 366 primary health care centers, including comprehensive health centers, health centers, health clinics and health posts. Additionally, there are about 700 private health facilities, consisting of private and non-profit, private for profit, and faith-based facilities.

Three LGAs were purposively selected to represent rural, semi-urban and urban areas within the state, and to align with the operations of donor-funded malaria programs. The selected areas included the state capital (urban and semi-urban) and Nkanu West (rural). In 2019, the malaria prevalence was estimated at 27.6% in Enugu state [18].

### Study design and sampling

We conducted household survey in the three LGAs to collect malaria service use, out-of-pocket payment, household assets and consumption. We conducted government, donors and malaria services documents review to collect the unit cost of different malaria services.

The sample size was determined using the estimated number of households per state which is approximately 1 million (an average household size of 5) [4]; a power of 80% probability of rejection of null hypothesis, 95% confidence level, 90% response rate and malaria prevalence level of 27.6% in southeast Nigeria [24]. A minimum sample of 516 households for the survey was estimated. Primary Healthcare (PHC) numbering system was used to determine households for the survey. The PHC numbering system aims at mapping or ascertaining the estimated coverage population of 10000 to 20000 per PHC, location of primary healthcare centres available in the country and also to determine how many are functional or otherwise [25]. In our study the first household that reported that it had malaria in the last one month before the survey or had received insecticide treated bed net in the last 6 months was randomly selected. Then other households were picked after every alternative household until the total number of respondents were collected across the selected LGAs.

### Data collection

Household survey was conducted between February 10-28, 2020 using interviewer-administered pre-tested structured questionnaires. Primary care giver or in his/her absence, the male head of each selected household was interviewed. Respondent in each household provided information on malaria prevention and treatment for the entire household members. The questionnaire was administered by trained field workers to the sample of randomly selected household members.

The household questionnaire has three sections: demographics, socioeconomic status and individual household health expenditure. Data were collected from respondents who have had malaria or have benefited from donor-funded malaria treatment services using a one-month

recall period for expenditures, except we used 6 months for bed nets. The one-month recall period was used to reduce the incidence of recall bias for a longer period.

Our study used benefit incidence analysis (BIA), which is method used to determine the distribution of benefits received by various socioeconomic groups using public health services for delivery of healthcare [26,27]. Malaria services in this study include outpatient care, malaria diagnosis, malaria treatment and malaria preventive measures including ownership and use of bed nets and in door residual house spraying services [28]. The unit costs of malaria services were collected from several sources, including (1) standard unit hospital charges for malaria in Enugu state; (2) unit costs for malaria services as set by the various donors, which represents the costs for treating outpatient malaria cases based on the international market prices; (3) cost data from the donors, which represents what is expected to be the national price for outpatient malaria services [29]; and (4) based on the cost data collected in #1-3, we computed unit costs for different malaria services [30], which present the value of what could be charged by health facilities that are involved in malaria treatment services. The study had to use the standardized hospital charges, costs from donors and international market prices because many of the respondents received malaria services (bed net, drugs etc) free, and we envisaged that respondents could provide biased information about real costs of malaria services.

We collected the medical and non-medical costs associated with malaria services in the study areas. Medical expenditures are costs incurred in the prevention or treatment of injury or disease. In this study, medical costs for malaria services include medical expenditure that were directly incurred by the patient, which include cost of registration, consultation, drugs, diagnostics and laboratory services. Non-medical costs on the other hand include additional costs in accessing healthcare such as transport and meals.

## Data analysis

We used STATA and SPSS software packages to analyze the data. The distributions of the key variables were analysed across socioeconomic status (SES) and rural-urban location. We used principal components analysis (PCA) to create a socioeconomic status (SES) index [31], based on information of the households' ownership of various assets such as radio, bicycle, motorcycle, tricycle, car, refrigerator, generator, kerosene lamp, together with the weekly household cost of food. Eigenvectors (weights) of the variables were derived from the correlation matrix to ensure that all data have equal weight in using the PCA to develop the index, which was then used to divide the households and individuals into five equal sized SES groups (quintiles (Q)): Q1 (most-poor); Q2 (poor); Q3 (average); Q4 (above average); and Q5 (least poor/rich) and was used to compare the sub-groups.

We analysed the use of different types of malaria services and then estimated the total healthcare costs. The total health service use per month include outpatient, diagnosis and treatment. The monthly OOP for the medical and non-medical costs associated with malaria services was also calculated and compared across different SES groups, and urban and rural dwellers. Costs were calculated in official local currency (Naira, N), then converted to US Dollar at the rate of N360.00 to one (1) US\$ in 2020. Kruskal-Wallis non-parametric test, which reports a $Chi^2$ statistic was used to compare differences in means of continuous variables.

The benefit incidence analysis (BIA) was done following the steps [32]: 1) Estimate the use of different types of the services from household survey; 2) Calculate the unit costs of each type of service using data from secondary sources; 3) Multiply the use by unit cost for each type of service for each population group (disaggregated by geographic location and socioeconomic group); 4) Deduct the direct user fee or out-of-pocket payments (from household survey) for each type of service for each population group/individual; 5) Aggregate the benefit

of use, expressed in monetary terms across different types of services for each individual/population group; 6) Compare the distribution of the malaria services benefits across the different population groups. Considering donors are major financing contributor to malaria services in Nigeria, we also estimated the benefit incidence of donor support by applying the proportion of malaria services funded by donors to the total benefits.

Concentration index was used to measure the level of equity in the distribution of benefits. Negative concentration index indicates the poor receive disproportionally more benefits ("pro-poor") while positive value shows the rich receive more benefits.

Data on the National DHS, 2018 and the study areas were used for comparisons of asset holding and living condition of the respondents in their different socioeconomic groups. Asset holding of the respondents were compared with the DHS data 2018. This was possible because questions on asset ownership were similar to the DHS data. The approach was to describe the SES of respondents in study sites with reference to the national population. Principal component analysis was used to compute weights for the ownership of identified list of assets in the 2018 DHS data. The weight scored were used to determine the scores for each SES.

### Ethical consideration

Ethical clearance was obtained from the University of Nigeria Teaching Hospital (UNTH) Ethics Committee and Duke University Campus Institutional Review Board, Protocol Number: 2019-0366. Written informed consent form was obtained and signed by all respondents. Primary care giver or in his/her absence, the male head of each selected household was interviewed. Data collectors were duly trained on handling confidential information from the respondents.

## Results

### Demographic and SES

We surveyed a total of 542 households (2795 individuals). Table 1 presents the demographic and socioeconomic characteristics of the respondents. The mean age of the respondents was 31.8 years for the combined sample, 33.7 years for the urban area, 31.3 years for the semi-urban area and 30.4 years for the rural areas. The majority of respondents in the urban, semi-urban and rural areas were females and married. Up to one-quarter of respondents in each study groups were engaged in big business (business with large capital) or were self-employed. More than half had secondary education. On average, each household consisted of 5.2 people, with about 1.5 of those being children under 5 years old.

### Use of malaria services

Among the surveyed households, 53.9% owned bed net and 31.6% used malaria drugs in the past months. RDTs were used by 7.9% of households, while lab services and IRS were used by only 4.6% and 1.7% of households, respectively. Urban populations had a higher proportion of households with and using bed nets, and receiving RDTs and lab services, compared to semi-urban and rural groups. However, rural households had the highest usage of malaria drugs, with 38.9% reporting usage, which was the highest among the three groups (Fig 1a). The least poor and the less poor quintiles (Q5&Q4) had relatively higher proportion of household with bed net and IRS usage, while the less poor and average quintiles (Q4&Q3) had higher proportions using RDTs, malaria drugs and lab services (Fig 1b). The concentration index for all services were positive, except for malaria drugs (p<0.01), indicating the use of malaria services was not pro-poor.

**Table 1. Demographic and socioeconomic characteristics of the respondents (N=542).**

| Variable | Combined | Urban | Semi-urban | Rural |
|---|---|---|---|---|
| | n (%) | n (%) | n (%) | n (%) |
| **Sample size** | 542(100) | 181(33.4) | 181(33.4) | 180(33.2) |
| **Age:** Mean (SD) | 31.8±7.1 | 33.7±7.0 | 31.3±7.9 | 30.4±5.8 |
| **Sex** | | | | |
| Male | 27 (5.0) | 23(12.7) | 2(2.0) | 2(1.1) |
| Female | 515(95.0) | 158(87.3) | 179(98.9) | 178(98.9) |
| **Marital Status** | | | | |
| Single | 20(3.7) | 1(0.6) | 16(8.8) | 3(1.7) |
| Married | 507(93.5) | 175(96.7) | 157(86.7) | 175(97.2) |
| Widowed/Divorced | 15(02.8) | 5 (2.8) | 8 (4.4) | 2 (1.1) |
| **Occupation** | | | | |
| Unemployed | 93(17.2) | 35(19.3) | 43(23.8) | 15(8.3) |
| Subsistence farming | 3(0.6) | 0(0.0) | 2(1.1) | 1(0.6) |
| Petty trading | 119(22.0) | 36(19.9) | 38(21.0) | 45(25.0) |
| Artisan | 79(14.6) | 24(13.3) | 19(10.5) | 36(20.0) |
| Civil servant | 45(8.3) | 26(14.4) | 7(3.9) | 12(6.7) |
| Private sector employee | 41(7.6) | 11(6.1) | 13(7.2) | 17(9.4) |
| Big business/self employed | 153(28.2) | 49(27.1) | 59(32.6) | 45(25.0) |
| Other (e.g., Clergy) | 9(1.7) | 0(0.0) | 0(0.0) | 9(5.0) |
| **Highest level of education** | | | | |
| No formal education | 3(0.6) | 0(0.0) | 2(1.1) | 1(0.6) |
| Primary school | 36(6.6) | 6(3.3) | 16(8.8) | 14(7.8) |
| Junior secondary | 49(9.0) | 8(4.4) | 19(10.5) | 22(12.2) |
| Senior Secondary | 316(58.3) | 96(53.0) | 112(61.9) | 1.8(60.0) |
| Polytechnic/University | 138(25.5) | 71(39.3) | 32(17.5) | 35(19.5) |
| **Number of people living in household** (Mean ±SD) | 2795 (5.2±1.7) | 920 (5.1±1.7) | 938 (5.2±1.6) | 937 (5.2±1.8) |
| Number of children aged < 5 year in a household | 785 | 268 | 247 | 270 |
| Number of pregnant women in a household | 76 | 20 | 26 | 30 |
| **Number of household member had malaria in the past 6 months** | 531(98.0) | 177(97.8) | 176(97.2) | 178(98.9) |
| **Social economic Status (SES) \*** | | | | |
| Q1(Most poor) | 115(21.2) | 38(21.0) | 41(22.7) | 36(20.0) |
| Q2 (Poor) | 101(18.7) | 33(18.2) | 32(17.7) | 36(20.0) |
| Q3 (Average) | 108(19.9) | 36(19.9) | 36(19.9) | 36(20.0) |
| Q4 (Above average) | 116(21.4) | 41(22.7) | 37(20.4) | 38(21.1) |
| Q5(Least poor) | 102(18.8) | 33(18.2) | 35(19.3) | 34(18.9) |

*The SES quintiles were calculated at the whole survey sample and then applied to respondents from urban, semi-urban and rural regions.*

Among the households that used the services, all paid OOP for malaria drugs, spray and lab services. Additionally, 88.4% of the households that used RDTs paid for them, and about 36.6% of household that owned bed net paid OOP for them. Fewer households in semi-urban and rural areas paid for bed net compared to those in urban areas. However, urban and rural households were more likely to pay for RDTs than those in semi-urban area (Fig 2a). Fewer households in the least and less poor quintiles (Q5&Q4) paid for bed nets, while the average and less poor quintiles (Q3&Q4) paid for RDTs (Fig 2b). The concentration index for proportion of OOP payments for malaria services was positive except for malaria drugs (p<0.01).

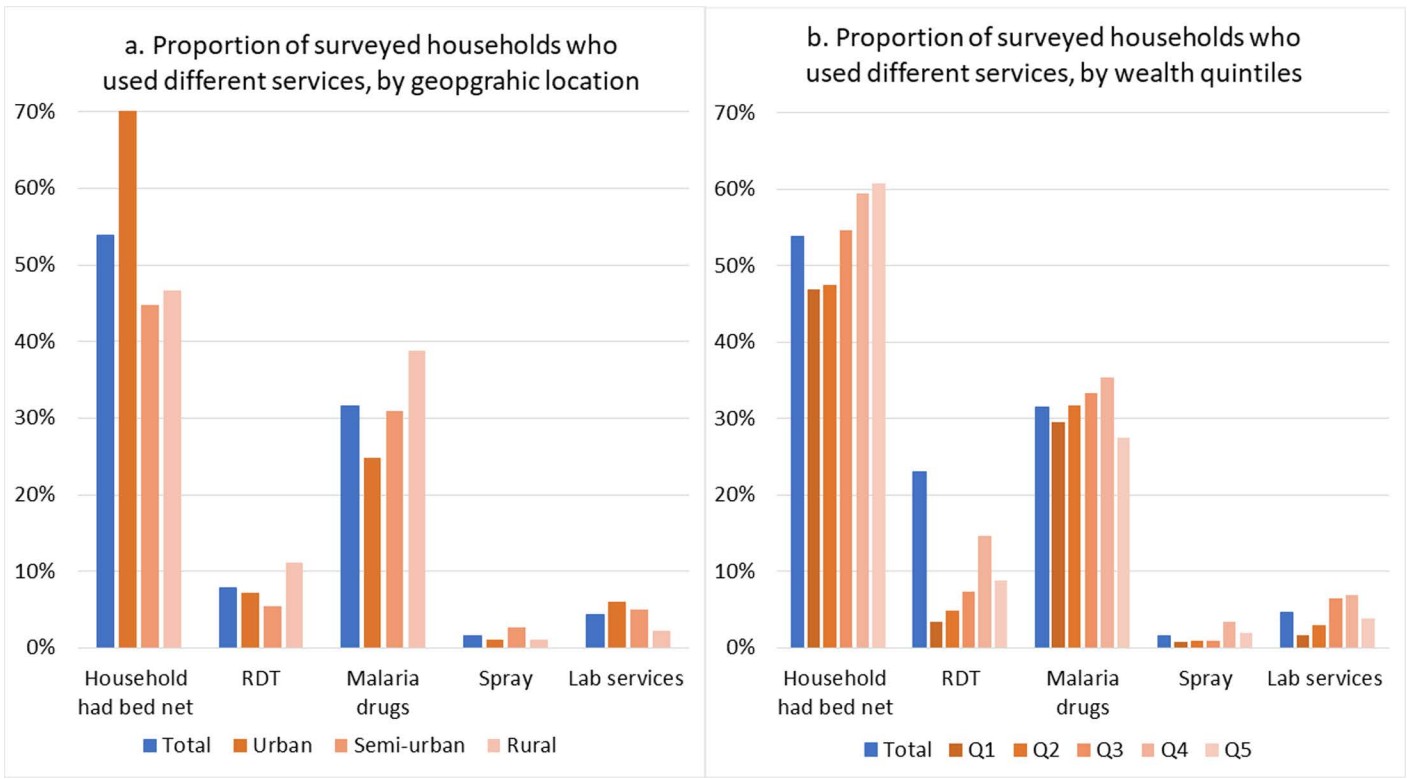

**Fig 1. Use of malaria services in the past month: a) by geographic location; b: by wealth quintiles (Q1(Most poor), Q2 (Poor); Q3 (Average); Q4 (Above average); Q5(Least poor)).**

The average total OOP expense per malaria case was $0.31 per household, with $0.29 allocated to medical cost and $0.02 for non-medical cost (Table 2). On average, rural households had the highest OOP for medical cost, non-medical cost and total OOP, followed by urban and semi-urban households. The least poor quintile (Q5) incurred the highest OOP, followed by the average quintile (Q3). After aggregating multiple malaria cases within the same household, the total OOP for malaria in the past month was $0.53 per household (Table 2). OOP for medical cost accounted for most of the malaria financial burden. Rural households had the highest OOP, followed by urban and semi-urban households. The least poor quintile had the highest OOP while the poorest quintile had the lowest (Table 2).

## Benefit incidence for malaria services

Based on the cost information collected from multiple sources, we computed the unit cost of different malaria services and estimates the gross benefit of these services. Table 3 presents the unit cost: $3.6 for bed net, $6.5 for RDT, $4.4 for malaria drugs, and $4.4 and $6.5 for and spray and lab services, respectively.

The total gross benefit for malaria services among the surveyed households was $1836.7, with $385.2, $247.0, $1008.9, $39.6 and $156.0 allocated to bed nets, RDTs, malaria drugs, spray and laboratory services, respectively. After deducting the total OOP medical costs from the gross benefit, the net benefit for malaria services was $1679.5. Donors funded 42.4% malaria services in Nigeria [33] contributing a benefit of $705.4. While both urban and rural households received about $659 in gross benefit from malaria services, urban households experienced a much higher net benefit ($632) compared to rural households ($532). The

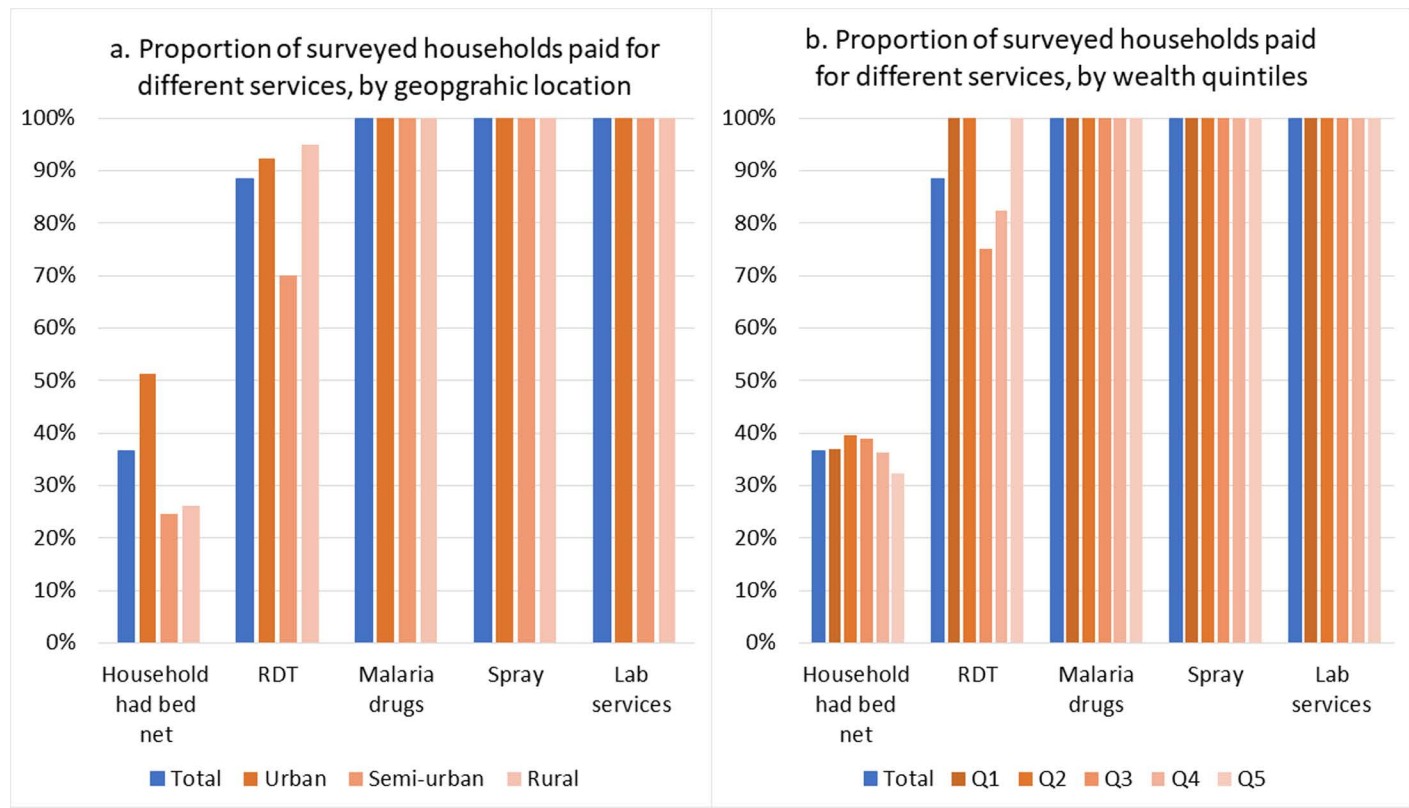

**Fig 2.   Proportion of households that paid for malaria services in the past month: a) by geographic location; b: by wealth quintiles (Q1(Most poor), Q2 (Poor); Q3 (Average); Q4 (Above average); Q5(Least poor)).**

**Table 2.  OOP for malaria services.**

| | Average OOP in the past month, US$, mean (SD) | | | Total OOP in the past month, US$, mean (SD) | | |
|---|---|---|---|---|---|---|
| | OOP for Medical cost | OOP for non-medical cost | Total OOP | OOP for Medical cost | OOP for non-medical cost | Total OOP |
| **Total (N=542)** | 0.29 (2.26) | 0.02(0.2) | 0.31(2.44) | 0.51(2.76) | 0.02(0.20) | 0.53(2.93) |
| **By location** | | | | | | |
| Urban (n=181,33.4%) | 0.14(1.17) | 0.01 (0.04) | 0.14(1.20) | 0.41(1.77) | 0.01(0.04) | 0.42(1.80) |
| Semi-urban (n=181,33.4%) | 0.06 (0.52) | 0.01 (0.09) | 0.07(0.60) | 0.18(0.92) | 0.01(0.09) | 0.19(0.98) |
| Rural (n=180,34.2%) | 0.66 (3.68) | 0.04 (0.33) | 0.70(3.98) | 0.93(4.32) | 0.04 (0.33) | 0.98(4.63) |
| **SES quintiles** | | | | | | |
| Q1 (n=115,21.2%) | 0.10(0.62) | 0.01 (0.11) | 0.12(0.72) | 0.17(0.90) | 0.01 (0.11) | 0.18(0.99) |
| Q2 (n=101,18.7%) | 0.22(1.25) | 0.01(0.05) | 0.23 (1.30) | 0.38(1.48) | 0.01(0.05) | 0.39(1.52) |
| Q3 (n=108, 19.9%) | 0.26(1.60) | 0.01(0.07) | 0.27 (1.66) | 0.47(1.82) | 0.01(0.07) | 0.48(1.87) |
| Q4 (n=116, 21.4%) | 0.16(1.37) | 0.01 (0.04) | 0.17(1.40) | 0.55 (2.14) | 0.01 (0.04) | 0.55(2.17) |
| Q5 (n=102,18.8%) | 0.73(4.49) | 0.06 (0.43) | 0.79 (4.90) | 1.00(5.34) | 0.06 (0.43) | 1.06(5.75) |

N360 = 1USDollar. Medical costs are direct costs for malaria related services including consultation, laboratory test, drugs etc while non-medical costs are the indirect costs such as transport and accommodation.

Table 3. Benefit for malaria services.

| Services | Unit Cost (₦) | ($) |
|---|---|---|
| Bed net | 1300.0 | 3.6 |
| RDT | 2349.0 | 6.5 |
| Malaria drugs | 2130.4 | 5.9 |
| Spray | 1588.9 | 4.4 |
| Lab services | 2331.7 | 6.5 |

Table 4. Benefit incidence for malaria services.

| | Gross benefit for each service ($) | | | | | | Total OOP for medical cost | Benefit for malaria services ($) | |
|---|---|---|---|---|---|---|---|---|---|
| | Bed net | RDT | Malaria drugs | SP | Lab services | Gross benefit | Net benefit | Donor benefit | |
| Total (n=542) | 385.2 | 247.0 | 1008.9 | 39.6 | 156.0 | 1836.7 | 157.2 | 1679.5 | 705.4 |
| By location | | | | | | | | | |
| Urban(n=181) | 234.0 | 78.0 | 265.5 | 8.8 | 71.5 | 657.8 | 25.3 | 632.5 | 265.7 |
| Semi-urban(n=181) | 72.0 | 45.5 | 330.4 | 22.0 | 58.5 | 528.4 | 10.9 | 517.5 | 217.4 |
| Rural(n=180) | 79.2 | 123.5 | 413.0 | 8.8 | 32.5 | 650.5 | 118.8 | 531.7 | 223.3 |
| P-value | 0.14 | 0.07 | 0.02 | 0.1 | 0.04 | 0.01 | 0.05 | 0.03 | 0.03 |
| SES quintiles | | | | | | | | | |
| Q1(115) | 72.0 | 26.0 | 200.6 | 4.4 | 13.0 | 316.0 | 11.5 | 304.5 | 127.9 |
| Q2(101) | 68.4 | 32.5 | 188.8 | 4.4 | 19.5 | 313.6 | 22.2 | 291.4 | 122.4 |
| Q3(n=108) | 82.8 | 39.0 | 212.4 | 4.4 | 45.5 | 384.1 | 28.1 | 356.0 | 149.5 |
| Q4(n=116) | 90.0 | 91.0 | 241.9 | 17.6 | 52.0 | 492.5 | 18.6 | 473.9 | 199.0 |
| Q5(n=102) | 72.0 | 58.5 | 165.2 | 8.8 | 26.0 | 330.5 | 74.5 | 256.0 | 107.5 |
| P-value | <0.01 | 0.02 | <0.01 | 0.04 | 0.01 | 0.05 | <0.01 | <0.01 | <0.01 |
| Concentration index | 0.06 | 0.42 | 0.001 | 0.46 | 0.31 | 0.11 | 0.65 | 0.06 | 0.06 |

concentration index was positive for both gross and net benefit, indicating the less poor households benefited more from malaria services (Table 4).

## Discussion

This study analysed the use, financial burden and benefit incidence of malaria services in Enugu, Nigeria. Among the survey respondents, over half of households owned a bed net, and about one-third used malaria drugs in the past months. All users have paid OOP for malaria drugs, sprays and lab services while over a third paid OOP for bed nets. Both the gross and net benefits for malaria bed nets and other services were pro-rich among the survey respondents.

About one-third of the surveyed households used malaria drugs in the past month. However, fewer than 15% households received RDTs or lab services, indicating that more than half of the malaria treatments were initiated without any diagnostic testing. Bed nets and sprays are preventive measures, while over half of the surveyed households owned bed nets, only 1.7% used sprays. Despite this, the overall use of malaria preventive approaches remains relatively low in Enugu, Nigeria. Urban households and those in the least poor quintiles used more bed nets, RDTs and lab services compared to their counterparts, highlighting inequitable access to malaria services across geographic locations and SES groups. To reduce the malaria incidence in Nigeria, substantial efforts are needed to improve the use of bed nets and sprays, as well as access to malaria diagnostics. Rural and poorer households also need greater attention.

Our study found that more urban residents used bed nets while more rural residents used malaria drugs. The study also revealed that wealthier households benefited from donor-funded bed nets than the poorer households. This finding aligns with studies in East Africa [34], which showed that donor-funded distribution of bed nets to targeted households benefited wealthier households more than poorer households in Tanzania and Uganda, but not in Angola.

The financial burden of malaria services may contribute to the underuse of malaria diagnostic services in Nigeria. According to the survey, all or most households paid OOP for malaria drugs, lab services, RDTs and sprays. Among malaria services, only bed nets have alleviated financial barriers for some users, as about two-thirds of the bed nets were distributed free of charge. However, relatively fewer households in the least and less poor quintiles (Q5&Q4) paid for bed nets, indicating the inequitable distribution of free bed net. To remove the financial burden of using malaria services, more resources and financial support should be allocated, particularly for poor households.

It is also important to note that this study found that rural residents spent more on total OOP ($0.98) than their urban counterparts ($0.42). This finding is consistent with other studies [35,36], which have found that urban residents are generally more informed, have better access to malaria services, and exhibit better health-seeking behavior. These factors enable them to demand care through more cost-effective options compared to rural residents.

The total gross benefit for malaria services among the surveyed households was $1836.7, with malaria drugs contributing the most, followed by bed nets. While treatment for malaria is essential, the diagnostic services such as RDTs and lab services, as well as preventive measures such as bed nets and sprays, also require greater attention and support. The less poor households benefited more from malaria services, underscoring the need for increased support to ensure that poorer households also benefit equitably.

Our study revealed inadequate use of malaria services, a heavy financial burden, and a pro-rich distribution of benefits. These findings are aligned with previous studies in Nigeria and sub-Saharan Africa [37,38,39,40], which highlight the significant financial burdens faced by households for malaria diagnosis and treatment.

Reports from Enugu State Ministry of Health from 2020 to 2023 indicate specific government budgets allocated for malaria treatment and control. The PMI also provides funding and support, not only for bed nets, but also for indoor residual spray and the delivery of malaria drugs to under 5 and pregnant women. However, such benefits, including funding for indoor residual spray or "free" malaria drugs, were not observed in this study. All households that used malaria drugs paid OOP, likely at a subsidized cost. Additionally, only 1.7% of surveyed households used the spray. This evidence suggests a need for increased promotion and funding for malaria services in Enugu State and Nigeria as a whole.

Considering that Nigeria currently receives financial support from donors and is in the process of transition away from this support, this study also estimated the donor benefit, which was found to be pro-rich. While the burden of malaria is evident across all socioeconomic groups examined in the study, poorer households are at higher risk of forgoing services due to financial barriers. The Nigeria government should mobilize more resources to support the provision of malaria services, ensuring that financial barriers do not prevent vulnerable households from accessing necessary care.

This study has several limitations. First, we did not include the use of inpatient services for malaria, as the response rate from the household survey on inpatient services use was too small. As a result, the burden of malaria care may have been underestimated. Second, although a one-month recall period is commonly used in major household surveys, we acknowledge that it may still be affected by recall bias. Third, the use of wealth index as a

measure of socioeconomic status is a static wealth measure and may not accurately reflect a household's struggle to pay for emergency malaria treatment services. Further research is needed to better understand households' financial coping strategies for seeking care and the role of existing malaria programs. Lastly, IPTp, a core components of PMI funding, was not included in the analysis due to a lack of data.

## Conclusion and policy implications

This study examined the use, financial burden and the benefit incidence of malaria services in Enugu state. The study found that financial spending and use of malaria services are not equitable. There is a clear need for improved access to affordable malaria prevention and treatment services in the state.

## Supporting information

**S1 Checklist. Checklist.**
(DOCX)

## Acknowledgements

We would like to thank the Enugu state Ministry of Health and the health facilities for giving us an approval to collect data from their facilities. We would also like to thank the respondents for their willingness and consent to participate in the study.

## Author contributions

**Conceptualization:** Eric Obikeze, Wenhui Mao, Obinna Onwujekwe.

**Formal analysis:** Eric Obikeze, Wenhui Mao, Uchenna Ezenwaka, Ifeyinwa Arize, Osondu Ogbuoji, Obinna Onwujekwe.

**Investigation:** Eric Obikeze, Uchenna Ezenwaka, Ifeyinwa Arize, Osondu Ogbuoji, Obinna Onwujekwe.

**Methodology:** Eric Obikeze, Wenhui Mao, Uchenna Ezenwaka, Osondu Ogbuoji, Obinna Onwujekwe.

**Supervision:** Uchenna Ezenwaka, Osondu Ogbuoji, Obinna Onwujekwe.

**Visualization:** Wenhui Mao.

**Writing – original draft:** Eric Obikeze.

**Writing – review & editing:** Wenhui Mao, Uchenna Ezenwaka, Ifeyinwa Arize, Osondu Ogbuoji, Obinna Onwujekwe.

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
