## [Decision Letter · Decision Letter 0]

10 Jul 2024

PGPH-D-24-00791

Who benefits from the donor supported malaria programme in Enugu State, Nigeria? A benefit incidence analysis

Dear Dr. Mao,

Thank you for submitting your manuscript to PLOS Global Public Health. After careful consideration, we feel that it has merit but does not fully meet PLOS Global Public Health’s publication criteria as it currently stands. Therefore, we invite you to submit a revised version of the manuscript that addresses the points raised during the review process.

Please note that we have only been able to secure a single reviewer to assess your manuscript. We are issuing a decision on your manuscript at this point to prevent further delays in the evaluation of your manuscript. Please be aware that the editor who handles your revised manuscript might find it necessary to invite additional reviewers to assess this work once the revised manuscript is submitted. However, we will aim to proceed on the basis of this single review if possible. 

The manuscript has been evaluated by one reviewer, and their comments are available below.

The reviewer has raised a number of concerns. Specifically, they have recommended that you carefully edit the manuscript, and add further details to the methods and results. Could you please carefully revise the manuscript to address all comments raised?

We look forward to receiving your revised manuscript.

Kind regards,

Johanna Pruller, Ph.D.

Staff Editor

Journal Requirements:

1. “Please include a complete copy of PLOS’ questionnaire on inclusivity in global research in your revised manuscript. Our policy for research in this area aims to improve transparency in the reporting of research performed outside of researchers’ own country or community. The policy applies to researchers who have travelled to a different country to conduct research, research with Indigenous populations or their lands, and research on cultural artefacts. The questionnaire can also be requested at the journal’s discretion for any other submissions, even if these conditions are not met.  Please find more information on the policy and a link to download a blank copy of the questionnaire here: https://journals.plos.org/globalpublichealth/s/best-practices-in-research-reporting. Please upload a completed version of your questionnaire as Supporting Information when you resubmit your manuscript.”"

2. "Please provide separate figure files in .tif or .eps format.

Additional Editor Comments (if provided):

Reviewers' comments:

Reviewer's Responses to Questions

**Comments to the Author**

1. Does this manuscript meet PLOS Global Public Health’s publication criteria ? Is the manuscript technically sound, and do the data support the conclusions? The manuscript must describe methodologically and ethically rigorous research with conclusions that are appropriately drawn based on the data presented.

Reviewer #1: Yes

2. Has the statistical analysis been performed appropriately and rigorously?

Reviewer #1: Yes

3. Have the authors made all data underlying the findings in their manuscript fully available (please refer to the Data Availability Statement at the start of the manuscript PDF file)?

Reviewer #1: No

4. Is the manuscript presented in an intelligible fashion and written in standard English?

Reviewer #1: Yes

5. Review Comments to the Author

Reviewer #1: I enjoyed reading this manuscript and I believe it addresses an important knowledge gap in malaria financing and control especially for Nigeria. I have few suggestions for the authors to consider.

1. I think the paper would benefit more with additional editing to make the sentences more succinct and clearer. For instance:

(a)Page 4 line 94-98 is a very long winding sentence that could be broken into two or three sentences for clarity.

(b)Page 6, line 137 "..based on the operational of donor funded malaria programs" Should this read as "based on the operations of donor funded malaria program?" The word "operational' seem like is being misused here.. please check and confirm.

(c) Page 10 line 231, I believe it was meant to read "SES of respondents in study sites" and not "sties"

There are few other areas that could benefit more with careful editorial review by the authors.

2. Given the broad definition of Benefits studies in financing and Economics, I suggest the authors describe clearly what type of benefits they are focusing on. To some readers, they may be thinking of benefits accrued from productivity work days avoided by not falling sick from malaria. I think it would be beneficial for authors to be clear upfront on how they intend to measure these benefits and why such approach is appropriate in this context.

3. Authors use words that are not defined or could be unclear to authors. For example on Page 10 the last sentence, line 247, '... each study groups were either in big business or..." It is unclear to the reader what 'big business' refers to here. Is it businesses with big capital, businesses in several states, what is exactly being meant when one says 'they belonged in big businesses?"

4. References need to be added in some parts where sentences are used loosely. For example page 6, line 139 will benefit if a reference is added to show where the information is coming from.

5. It is unclear what authors are referring to when on page 6. line 143-144 they say .."We conducted document review to collect the unit cost of different malaria services." What documents are being referred to here? Government documents? Peer reviewed documents that are publicly available and accessible? Clarity would help readers understand what type of documents are being referred to here.

6. Page 11 line 251-253, I do not think the sentence indicating that the survey sample were evenly distributed among the five quintiles of the NDHS 2018 adds value since the whole point is to have these groups evenly distributed, unless I am missing something here or the claim being made is incomplete. Please, check.

7. There is interesting result showing that rural households paid more than twice OOP compared to urban households ($098 vs. $0.42 - See Table 2 on page 13). There is no discussion of why this is the case. Most of this is also shown to be on medical costs, why is this being the case? Self treatment and purchase of medicines at drug stores? more discussion on possible reasons for this spending by rural poor households would surely be warranted here.

8. A definition of what medical costs includes would also be useful. Does this include drugs? consultation fees, bed net purchase, RDT Kits? laboratory tests? It is unclear since these are also listed separately in some instances.

9. These findings in some way corroborates with other finding published from Sub Saharan Africa. For instance Njau JD et al (2013) "Exploring the impact of targeted distribution of free bed-nets on household bed-net ownership, socioeconomic disparities and childhood malaria infection rates: Analysis of National Malaria Survey data from three sub Saharan African Countries" found that donor funded distribution of bed-nets to targeted households benefited wealthier households than poorer households in Tanzania and Uganda, but not in Angola. All three countries were funded by the President Malaria Initiative (PMI). It would be nice for authors to review the paper and see what similarities they can draw from the paper in their discussions section.

10. The authors should also mention the limitation of Asset index in their study limitation section. The Asset Index is known to be a static wealth measure and is not a good reflection of households struggling for cash to pay for emergency malaria treatment service needs. For this reason it is important for readers to know the mechanics of Wealth Index measure and what it aims to achieve in these types of studies.

11. I also think they should mention the fact that the study does not include the use or uptake of Intermittent Preventive Treatment of Malaria in pregnant women which is one of the core components of PMI funding. IPTp is an important malaria control strategy funded through PMI and the fact that the benefits of this strategy are not being reported here is a limitation that is worth discussing at least briefly.

6. PLOS authors have the option to publish the peer review history of their article (what does this mean? ). If published, this will include your full peer review and any attached files.

**Do you want your identity to be public for this peer review?** For information about this choice, including consent withdrawal, please see our Privacy Policy .

Reviewer #1: **Yes: ** Joseph D. Njau

---

## [Decision Letter · Decision Letter 1]

15 Jan 2025

PGPH-D-24-00791R1

Who benefits from the donor supported malaria programme in Enugu State, Nigeria? A benefit incidence analysis

Dear Dr. Mao-

Thank you for submitting your manuscript to PLOS Global Public Health. We are pleased to see the changes you have brought about in view of the reviewers' comments. After careful consideration, we feel that the manuscript may benefit from strong language editing to completely meet  PLOS Global Public Health’s publication criteria. Therefore, we invite you to submit a revised version of the manuscript that addresses the points raised during the review process.

We look forward to receiving your revised manuscript.

Kind regards,

Shifa S. Habib

Academic Editor

Journal Requirements:

Additional Editor Comments (if provided):

Reviewers' comments:

Reviewer's Responses to Questions

**Comments to the Author**

1. If the authors have adequately addressed your comments raised in a previous round of review and you feel that this manuscript is now acceptable for publication, you may indicate that here to bypass the “Comments to the Author” section, enter your conflict of interest statement in the “Confidential to Editor” section, and submit your "Accept" recommendation.

Reviewer #1: All comments have been addressed

2. Does this manuscript meet PLOS Global Public Health’s publication criteria ? Is the manuscript technically sound, and do the data support the conclusions? The manuscript must describe methodologically and ethically rigorous research with conclusions that are appropriately drawn based on the data presented.

Reviewer #1: Yes

3. Has the statistical analysis been performed appropriately and rigorously?

Reviewer #1: Yes

4. Have the authors made all data underlying the findings in their manuscript fully available (please refer to the Data Availability Statement at the start of the manuscript PDF file)?

Reviewer #1: Yes

5. Is the manuscript presented in an intelligible fashion and written in standard English?

Reviewer #1: Yes

6. Review Comments to the Author

Reviewer #1: The authors have done commendable job addressing the issues I suggested to be addressed in their previous version. I am confident that the paper reads better and has improved significantly.

However, I think the paper will still benefit with some editing. I suggest if possible the authors consider to send their paper to a professional language editor to help improve the quality of the paper. For instance on page 19, line 422 "..Therefore, we could have under estimate the burden of malaria care." I believe "underestimate" should be one world and not two words. The sentence will also read better if it is written as follows ".Therefore we may have underestimated the burden of malaria care." There are other instances where wording and sentences could be improved for better clarity. Otherwise, it is a job well done and I commend the paper for publication.

7. PLOS authors have the option to publish the peer review history of their article (what does this mean? ). If published, this will include your full peer review and any attached files.

**Do you want your identity to be public for this peer review?** For information about this choice, including consent withdrawal, please see our Privacy Policy .

Reviewer #1: **Yes: ** JD Njau

---

## [Editor Report · Decision Letter 2]

27 Jan 2025

Who benefits from the donor-supported malaria programme in Enugu State, Nigeria? A benefit incidence analysis

PGPH-D-24-00791R2

Dear Dr. Mao,

We are pleased to inform you that your manuscript 'Who benefits from the donor-supported malaria programme in Enugu State, Nigeria? A benefit incidence analysis' has been provisionally accepted for publication in PLOS Global Public Health.

Best regards,

Shifa S. Habib

Academic Editor